# *Trichoderma asperellum* Secreted 6-Pentyl-α-Pyrone to Control *Magnaporthiopsis maydis*, the Maize Late Wilt Disease Agent

**DOI:** 10.3390/biology10090897

**Published:** 2021-09-11

**Authors:** Ofir Degani, Soliman Khatib, Paz Becher, Asaf Gordani, Raviv Harris

**Affiliations:** 1Migal—Galilee Research Institute, Tarshish 2, Kiryat Shmona 11016, Israel; solimankh@migal.org.il (S.K.); pazbec@gmail.com (P.B.); asigordani1@gmail.com (A.G.); ravivharris@gmail.com (R.H.); 2Faculty of Sciences, Tel-Hai College, Upper Galilee, Tel-Hai 12210, Israel

**Keywords:** biological control, *Cephalosporium maydis*, chromatography, crop protection, fungus, *Harpophora maydis*, microflora, mass spectrometry

## Abstract

**Simple Summary:**

The maize (*Zea mays* L.) late wilt disease, caused by the fungus *Magnaporthiopsis maydis,* is considered the most severe threat to commercial maize production in Israel and Egypt. Various control strategies have been inspected over the years. The current scientific effort is focusing on eco-friendly approaches against the disease. The genus *Trichoderma*, a filamentous soil and plant root-associated fungi, is one of the essential biocontrol species, demonstrating over 60% of all the listed biocontrol agents used to reduce plant infectious diseases. They produce different enzymes and elicit defense responses in plants, playing a significant role in biotic and abiotic stress tolerance, hyphal growth, and plant growth promotion. *Trichoderma asperellum* was found to have biocontrol ability and protect crops against various plant pathogenic fungi, including the maize late wilt disease causal agent. This research aimed at isolating and identifying *T. asperellum* secondary metabolites with antifungal action against *M. maydis*. From *T. asperellum* growth medium, the 6-Pentyl-α-pyrone secondary metabolite was isolated and identified with high potent antifungal activity against *M. maydis*. This compound previously exhibited antifungal activities towards several plant pathogenic fungi. Achieving clean and identified *T. asperellum* active ingredient(s) secreted product(s) is the first step in revealing their commercial potential as new fungicides. Follow-up studies should test this component against the LWD pathogen in potted sprouts and the field.

**Abstract:**

Late wilt disease (LWD) is a destructive vascular disease of maize (*Zea mays* L.) caused by the fungus *Magnaporthiopsis maydis.* Restricting the disease, which is a significant threat to commercial production in Israel, Egypt, Spain, India, and other countries, is an urgent need. In the past three years, we scanned nine *Trichoderma* spp. isolates as biological control candidates against *M. maydis*. Three of these isolates showed promising results. In vitro assays, seedlings pathogenicity trials, and field experiments all support the bio-control potential of these isolates (or their secretions). Here, a dedicated effort led to the isolation and identification of an active ingredient in the growth medium of *Trichoderma asperellum* (P1) with antifungal activity against *M. maydis*. This *Trichoderma* species is an endophyte isolated from LWD-susceptible maize seeds. From the chloroform extract of this fungal medium, we isolated a powerful (approx. 400 mg/L) active ingredient capable of fully inhibiting *M. maydis* growth. Additional purification using liquid chromatography–mass spectrometry (LC–MS) and gas chromatography–mass spectrometry (GC–MS) separation steps enabled identifying the active ingredient as 6-Pentyl-α-pyrone. This compound is a potential fungicide with high efficiency against the LWD causal agent.

## 1. Introduction

More maize (*Zea mays* L.) is produced annually than any other grain, reflecting its importance in the global market. Maize late wilt disease (LWD) causes severe damage to cornfields throughout Israel. The disease is characterized by the rapid wilt of sweet and fodder maize, mainly from the tasseling stage until shortly before maturity [1,2,3]. The causal agent is the fungus *Magnaporthiopsis maydis*, recognized by two additional synonyms, *Cephalosporium maydis* and *Harpophora maydis* [4]. The pathogen is a soil-borne hemibiotroph [5] seed-borne [6] and is spread as spores, sclerotia, or hyphae on plants’ remains. *M. maydis* can live in the soil for long periods or by developing inside different host plants, such as *Lupinus termis* L. (lupine) [7], *Gossypium hirsutum* L. (cotton), *Citrullus lanatus* (watermelon), and *Setaria viridis* (green foxtail) [8].

The disease mode in susceptible maize genotypes is well-detailed in the literature. According to Sabet et al. [9], the infection occurs during the first three weeks of maize growth. After penetrating the roots, the fungus first appears in the xylem 21 days after seeding and spreads upwards. When tassels first emerged (ca. day 60), the fungus appeared throughout the stalk, the pathogen DNA levels peaked in the stems [10], and the first above-ground symptoms appeared shortly thereafter. About 10 days before harvest, the fungus hyphae and secreted materials block the plant’s water supply and lead to rapid dehydration and death. The symptoms are enhanced under drought conditions [11,12,13,14,15]. LWD may result in 100% infection and total yield loss in heavily infested fields planted with sensitive maize hybrids [1,16]. A parallel infection mode (with some delay) occurs in resistant cultivars, and the pathogen can infect the seeds of these non-symptomatic plants and enhance their spread [1,10].

Today, there is an urgent need to develop innovative approaches to study *M. maydis*, monitor its spread, and restrain its damage. LWD is regarded as the most harmful maize disease in commercial fields in Egypt [17] and Israel [18], poses a major threat in India and Spain, and is a serious concern to other countries [19,20,21]. Efforts have been made since its discovery in the 1960s [22] to restrict the disease using agricultural (balanced soil fertility and flood fallowing) [13,23], biological [24,25,26,27,28,29,30,31], physical (solar heating) [32], plant chemical [33], and fungicide options [1,34,35], with different degrees of success. Recently, for the first time since the discovery of LWD in Israel, an economical and efficient applicable solution was approved that can now be used on a large scale to protect susceptible maize varieties [36]. Notwithstanding this recent encouraging success in developing control strategies against LWD, the most common way to control the disease is by using highly resistant maize genotypes [2,37,38,39,40]. This approach is environmentally friendly, efficient, and cost-effective.

In Israel, for example, the amount of corn crops in metric ton yield per metric hectare exhibits a continuous upward tendency, from 17.5 in 1987–1996 to 20.1 in 2007–2016 (Israel Organization of Crops and Vegetables’ data). Effective risk management of LWD, primarily by avoiding sensitive maize cultivars’ growth, may contribute to this positive trend. Still, the highly aggressive isolates of *M. maydis* discovered in recent years [18,41,42,43] may pose a threat to resistant maize cultivars, which, after extensive cultivation for long periods, can lose their immunity. This scenario has taken place in Israel regarding the relatively resistant maize cultivar, Royalty, which became the leading maize cultivar during the late wilt disease outbreak in the 1990s [16,18] and gradually became increasingly more sensitive to LWD. Thus, this alarming situation is forcing researchers to seek alternative methods.

The chemical option also suffers from drawbacks. It was shown that the pathogen is present in the host tissues of successfully chemically treated plants [3]. This finding hints at the potential risk that the pathogen could develop immunity to fungicides. Indeed, azoxystrobin-based treatment, the most effective antifungal compound against the late wilt pathogen [1,36], is under such threat. The rapid appearance of resistance to azoxystrobin and the consequential control failure in many crops has become increasingly problematic (summarized in [44]). Furthermore, the extensive use of fungicides poses a serious concern—their residual effects and toxicity may adversely affect the environment [45] and human health [46]. 

*Trichoderma* spp. and other microorganism-based biological controls of LWD have already been demonstrated in several studies (most recently [24,29]). Specifically, one of the most explored methods is the use of rhizobacteria as a biological barrier against LWD. Such a solution may assist in plant growth promotion and improve plant health (summarized by [29]). Alternatively, members of the fungal *Trichoderma* genus have high potential towards the same goal. Some *Trichoderma* species can form mutualistic endophytic relationships with several plant species [47], whereas others have biocontrol capability against fungal phytopathogens [48]. It was recently shown [24] that extracts of the microalgae *Chlorella vulgaris* with the *Trichoderma* species *T. virens*, or *T. koningii* were effective treatments against LWD under greenhouse and field conditions. These combinations resulted in a 72% reduction in disease incidence in the greenhouse and a 2.5-fold higher grain production under field conditions. Still, this method should be examined against the Israeli variants of the pathogen, and the action mechanism should be explored.

In Israel, corn is a significant crop in open areas, with a 3350 ha harvested area and volume production of 77,801 tons (FAOSTAT, 2019 Food and Agriculture Commodity Production data). Continued development of green solutions to efficiently control the LWD pathogen for commercial grain production and maize seed production is an urgent need. To this end, we recently identified three *Trichoderma* isolates with inhibitory activity against *M. maydis*: *T. longibrachiatum* (T.7407 from marine source [49]); *T. asperelloides* (T.203); and *T. asperellum* (P1), an endophyte isolated in our laboratory from maize seeds of a strain susceptible to LWD [27]. These isolates restricted the pathogen’s growth in culture plates, significantly reducing its establishment and development in seedlings’ corn plant tissues and considerably improving growth and crop indices in potted plants under field conditions [26]. Follow-up work was conducted to establish these isolates’ bioprotective potential for commercial corn production [25]. *T. asperellum* significantly rescues the infected sprouts’ growth indices and reduces *M. maydis* DNA in their roots [27]. This endophytic species treatment also excels in the field over a whole growth period [25], and at the season’s end, resulted in 1.6- and 1.3-fold improvements in lower stem and cob symptoms, respectively. Moreover, this treatment led to 4.9-fold lower *M. maydis* DNA levels in the plants.

The current research aimed at purifying and identifying ingredient(s) in the *T. asperellum*-secreted metabolites with antifungal activity against the LWD pathogen. To this end, we screened various *Trichoderma* isolates against the pathogen in liquid and solid growth media. We measured the dry weight parameter of the target fungus and colony diameter, respectively. The secreted metabolites of *T. asperellum* (which show high antifungal properties against the LWD pathogen) were subjected to active ingredient isolation and identification using liquid and gas chromatography–mass spectrometry (LC–MS and GC–MS). Finally, the purified active ingredient was tested in *M. maydis* cultures dose-dependent growth inhibition assay and in maize seeds phytotoxicity assay.

## 2. Materials and Methods

### 2.1. Fungal Species and Growth Conditions

The *M. maydis* isolate *Hm-2* (CBS 133165, CBS-KNAW Fungal Biodiversity Center, Utrecht, The Netherlands) was isolated by Prof. Amir Sharon (Tel Aviv University, Israel) from diseased maize plants collected from Sde Nehemia (Hula Valley, Upper Galilee, northern Israel) in 2001 and identified as previously described [10]. The fungus was grown on solid, rich potato dextrose agar (PDA) (Difco, Detroit, MI, USA) medium in the dark at 28 ± 1 °C for 4–7 days before being used. Liquid substrate growth was performed by sowing 10 fungal discs in an Erlenmeyer flask containing 150 mL potato dextrose broth (PDB, Difco Laboratories Detroit, Michigan, USA). The flasks were plugged with a breathable stopper, incubated for six days, and shaken at 150 rpm at 28 ± 1 °C in the dark.

The four *Trichoderma* spp. examined in this research were obtained from different sources (Table 1). *T. longibrachiatum* (isolates T7407 and T7507) separated from Mediterranean sponge *Psammocinia* sp. and the well-established biocontrol strain T203 (*T. asperelloides* [50]) were received courtesy of Prof. Oded Yarden (Hebrew University of Jerusalem, Israel) and were previously characterized [49]. The *Trichoderma asperellum* (P1) is an endophyte recovered from Prelude cv. (sweet maize from SRS Snowy River seeds, Australia, supplied by Green 2000 Ltd., Israel) grains, and identified as previously described [27]. All *Trichoderma* isolates were previously tested in the lab, and sprouts were selected for this study based on their high achievements in those tests (their intense bioprotective activity against *M. maydis*) [26,27]. The *Trichoderma* isolates’ growth conditions were similar to the *M. maydis* growth conditions described above unless otherwise indicated.

### 2.2. Effect of Trichoderma spp. Secreted Metabolites on M. maydis Cultures

The impact of *Trichoderma* isolate-secreted metabolites on *M. maydis* growth was evaluated in liquid (PDB) and solid (PDA) growth media. Each treatment was performed in five repetitions (unless otherwise indicated); each experiment was repeated twice, obtaining similar results. Five 6 mm mycelial discs were taken from the margins of 2-day-old colonies of selected *Trichoderma* isolates (Table 1) and incubated in 150 mL rich liquid (PDB) substrate at a temperature of 28 ± 1 °C in the dark at 150 rpm shaking for six days. The growth medium of the liquid PDB *Trichoderma* spp. cultures were separated by filtration using a Buchner funnel and Whatman filter no. 3. The growth medium pH was measured and adjusted to 5.1 ± 0.2 (the pH of PDB medium) with NaOH. The liquid was filtered again using biofilter bottles (0.22-micron filter, BIOFIL 500 mL vacuum bottle filter, Indore, India) for sterilization. From the filtered liquid, 100 mL was poured into a sterile 250 mL Erlenmeyer bottle. A sterilized 6% glucose solution was added to the liquid to a final concentration in a 2% bottle, identical to the amount of glucose in a standard PDB substrate. The control is PDB medium *M. maydis* cultures, maintained under the same conditions. Five colony agar disks of *M. maydis* were added to each Erlenmeyer bottle, and the flasks were plugged with a breathable stopper and incubated at 150 rpm at 28 ± 1 °C in the dark. After six days, the fungus’ mycelium was separated by filtration, and the dry biomass was determined after drying at 65 °C for 62 h.

In addition, the effect of the *Trichoderma* isolate-secreted products on solid media plates was examined. To this end, the *Trichoderma* cultures’ filtrate described above was used to prepare the PDA plates (instead of DDW). PDA powder was added to the filtrate according to the manufacturer’s recommendation. The mixture, after autoclave sterilization, was poured into Petri plates, and the plates were seeded with a 6 mm diameter *M. maydis* mycelial disc (from the margin of a fungus colony grown as described in Section 2.1). The colony growth rate was measured after six days compared to the growth rate of the fungus grown on a standard PDA substrate.

### 2.3. Purification and Identification of the Active Ingredients in the Secretion of T. asperellum

The *T. asperellum* growth medium (crude component), prepared as described in Section 2.2, was used for the active ingredients’ purification and identification. The resulting *Trichoderma* isolate growth fluid was lyophilized, and the precipitate was separated into fractions and treated using the following steps:A total of 100 mL of the supernatant was extracted twice with 100 mL chloroform.The extracted fractions were combined and evaporated entirely to separate them from the organic solvent.The chloroform extracts were tested for inhibitory activity against *M. maydis* by using them (instead of DDW) to prepare PDA solid substrate plates. Growth conditions were as described in Section 2.1.

The chloroform extracts had strong *M. maydis* antagonistic characteristics and were analyzed chemically to identify their active ingredients. A pure chromatography system, C-815 flash from BUCHI-Switzerland, connected to UV and an evaporative light-scattering detector (ELSD) was used to purify the active compounds with a silica gel column. The molecular structure elucidation of the active compound(s) was carried out using sets of analytical instrumentation.

#### 2.3.1. GC–MS Analysis

GC–MS analysis was performed using an Agilent 7890c gas chromatograph with a 5975C mass selective detector (GC–MSD). The system was equipped with a fused silica capillary column (Agilent J&W GC column HP5, length 30 m, diam. 0.32 mm, and a film thickness of 0.25 mm); the carrier gas was helium at 10 Psi, the split ratio was 10:1, the injector temperature was set to 250 °C, and the transfer liner was set to 280 °C. The column oven temperature program was started at 80 °C and ramped to 325 °C at 5 °C/min then kept at 325 °C for 10 min. The total run time was 59 min.

#### 2.3.2. HPLC Analysis

The samples were analyzed by injecting 10 μL of the chromatographed fraction solutions (2 mg/mL in methanol) into a UHPLC connected to a photodiode array detector (Dionex Ultimate 3000) with a reverse-phase column (Phenomenex RP-18, 150 × 4.0 mm, 3 μm). The mobile phase consisted of (A) DDW with 0.1% formic acid and (B) acetonitrile containing 1% formic acid. The gradient started with 0% B for 5 min, then increased to 98% B for 30 min, and maintained at 98% B for another 5 min at a flow rate of 1 mL/min.

#### 2.3.3. LC–MS/MS Analysis

The LC–MS/MS analysis was generated using a Q Exactive™ plus Hybrid Quadrupole-Orbitrap mass spectrometer (Thermo Fisher Scientific) equipped with a heated electrospray ionization source; UPLC Ultimate 3000 (Thermo Fisher Scientific) ESI capillary voltage was set to 3500 V, capillary temperature to 300 °C, gas temperature to 350 °C, and gas flow to 10 mL/min. The mass spectra (*m*/*z* 100–1000) were acquired in positive-ion mode (ESI^+^). Data processing was generated using Xcalibur™ and FreeStyle 1.6 software and MzCloud MS/MS database from Thermo Scientific™.

### 2.4. Seeds Phytotoxicity Assay

The susceptible cultivar of sweet maize Prelude cv. was chosen for the seed phytotoxicity test. This cultivar was tested previously and proved to be highly susceptible to late wilt [1,3]. To evaluate the secreted metabolite 6-Pentyl-α-pyrone (6-PP, synonyms include 6PAP, 6-n-pentyl-2H-pyran-2-one, 6-amyl-alpha-pyrone) phytotoxicity, we conducted a seed germination assay and compared the growth medium extract of *T. asperellum* (P1) to a commercial 6-PP compound (CAS Number: 27593-23-3, Sigma- Aldrich, Rehovot, Israel). The test included two steps. First, a plate inhibition assay was used as a bioassay to determine the antifungal activity of the P1 crude and the commercial 6-PP against *M. maydis*. To this end, the growth medium of P1 or commercial 6-PP compound was embedded in PDA at the indicated dosages and tested for *M. maydis* inhibition after three days, as described in Section 2.2.

Second, selected concentrations of the commercial 6-PP were tested in a seed germination assay against the P1 crude and a control. Five seeds of the Prelude maize cultivar (not treated with traditional seed treatment fungicides or insecticides) were dipped in 1% (*v*/*v*) sodium hypochlorite for 3 min, washed twice in sterile double-distilled water (DDW), and then placed in a Petri dish with 3.5 mL of the inspected solution dissolved to the desired concentration in autoclaved DDW. Then, five maize seeds were added to each Petri dish, and the plates were incubated at 28 °C in the dark. Untreated seeds (seeds soaked only in sterile tap water) were used as negative controls. Seed germination percentage determination of all the seeds in each treatment was conducted three days later. A germinating seed was defined as a seed in which the seed coat was broken by the radicle. 

## 3. Results

### 3.1. The Effect of Trichoderma spp. Secreted Metabolites on M. maydis Cultures

This study focused on three *Trichoderma* isolates (Table 1) that had previously exhibited antagonistic activity against *M. maydis* (the cause of LWD in corn). The *Trichoderma* spp.-secreted metabolite activity was tested against two positive controls: a *Trichoderma* isolate with no known inhibitory activity against *M. maydis* (T.7507) and *M. maydis* growth medium (PDB/PDA) without the presence of secretory products of *Trichoderma* app. In submerged cultures, we performed two consecutive experiments (with similar results) to verify the inhibitory activity of selected *Trichoderma* isolates. The results shown in Figure 1 reveal that two *Trichoderma* isolates, *T. asperellum* (P1) and *T. longibrachiatum* (T7407), significantly inhibit fungal growth (dry weight, *p* < 0.05) compared to the control—*Trichoderma*-free PDB substrate. Isolate T7407 results were significantly better than isolate T7507 (which has no known *M. maydis* inhibitory effect), although they both belong to the same species (*T. longibrachiatum*). *T.*
*asperelloides* (T203) had no apparent influence on *M. maydis* activity in this assay.

Two solid medium experiments followed the above experiment. These experiments yielded similar results, which were also similar to the submerged culture assay described above. The results obtained show that the growth fluid of the three *Trichoderma* isolates, T203 and T7407, P1, caused a significant (*p* < 0.05) decrease in *M. maydis* development (Figure 2). 

### 3.2. Purification and Identification of Active Ingredients in the Secretion of T. asperellum

Both *T. asperellum* (P1) and *T. longibrachiatum* (T7407) excel in restricting the LWD pathogen in the above tests (Figure 1 and Figure 2) and in the field [25]. Yet, *T. asperellum* (P1) is an endophyte isolated from grains of a maize genotype susceptible to LWD [27]. Thus, it is more likely that its secreted metabolites will have no phytotoxicity on plants. For this reason, it was selected for the subsequent analytic work aimed at isolating and identifying the active ingredient(s) in its growth medium. The active compound(s) in the growth medium of *T. asperellum* were extracted using chloroform. The extracts’ fraction was purified using a flash chromatography system equipped with a flash silica column, diode array, and ELSD detectors (Figure 3). From the growth medium of P1, we isolated a strong inhibitory chloroform extract (about 400 mg/liter) that can completely inhibit (at a concentration of 8 mg per ml) the growth of *M. maydis* (Figure 4). A gradient decrease in the *M. maydis* colony diameter was correlated to the gradually increasing amounts (in mg) of active ingredient(s) embedded in the PDA medium.

The *T. asperellum* (P1) growth medium extract underwent further cleaning and separation stages using flash direst phase chromatography to obtain a purified active compound (Figure 3A and Figure 4). Using LC–MS/MS and GC–MS, the compound was identified as 6-Pentyl-α-pyrone (6-PP, Figure 5 and Figure 6). This analysis revealed a sole active ingredient in the *T. asperellum*-secreted metabolites exhibiting potent antifungal activity against the LWD pathogen, *M. maydis.* Finally, the ingredient identification as 6-PP was proven using a standard.

To evaluate the secreted metabolite 6-PP phytotoxicity, we conducted a seed germination assay and compared the growth medium extract of *T. asperellum* (P1) to a commercial 6-PP compound. The test included two steps. First, a plate inhibition assay (Figure 7A) was used as a bioassay to determine the antifungal activity of the P1 crude and the commercial 6-PP against *M. maydis*. Fifty percent dilution of the P1 secreted metabolites had similar antifungal activity as the commercial compound’s 0.02% (*v*/*v*) concentration. Second, selected concentrations of the commercial 6-PP were tested in seed germination assay (Figure 7B) against the P1 crude. This assay revealed that the clean 6-PP (0.04%) was more phytotoxic than the crude (100%), which according to the plate bioassay, was expected to have similar antifungal activity. Still, at a relatively high concentration of clean 6-PP, the phytotoxicity at the sprouting phase was mild (36% inhibition compared to the control, *p* < 0.05).

## 4. Discussion

Maize late wilt disease is a destructive threat to commercial production and a constant concern to growers in highly infected areas, especially in Israel [25], Egypt [2,51], Spain, Portugal [28], and India [52,53]. Over the years, various control strategies have been suggested and tested (see details in the Introduction). Still, eco-friendly biological approaches are most desired since they support sustainability—a long-lived and healthy environment. In particular, these approaches are in line with the current trend of reducing the use of pesticides [54]. In recent years, several such green solutions to LWD were developed and approved in the field (most recently [25,26,28,29,37]). Yet, there is an urgent need to expand the range of available options and identify new environmentally friendly ways to eradicate the disease. The current work contributes to this effort and points to a compound from a biological source (*T. asperellum*) as a potential fungicide with high efficiency against the LWD causal agent.

*T. asperellum* (P1) is an endophyte that was isolated in our laboratory from corn seeds of a strain susceptible to LWD [27]. It was previously tested in the field and gained encouraging results in repressing *M. maydis* spread within the host plants and recovering the plants’ growth indices [25]. This species was discovered as being bioprotective against other phytopathogenic fungi. The use of *T. asperellum* to control various plant diseases includes the following recent examples: *Fusarium* wilt in *Stevia rebaudiana* [55], *Pratylenchus brachyurus* in soybeans [56], and pearl millet downy mildew caused by *Sclerospora graminicola* [57].

Based on literature reports, this strain presents four important functions that are beneficial for agricultural production. The first is growth promotion: *T. asperellum* (isolate SM-12F1) can improve the activities of soil enzymes associated with nutrient activation and antioxidant stress [58] and induce plant growth-promoting attributes of 1-aminocyclopropane-1-carboxylate deaminase, auxin, and siderophore production [59]. *T. asperellum*’s second important function is induced systemic resistance (ISR) in plants by a mechanism that employs ethylene and jasmonic acid signal transduction pathways [60,61]. The third important function is biocontrol, in which the *Trichoderma* species reduces the pathogens surrounding the plant roots by competing for nutrients and space (taking advantage of their rapid growth), inhibiting pathogen growth by mycoparasitism and secondary metabolite production [62]. The fourth important function is bioremediation, in which *T. asperellum* (SM-12F1) can remediate As-contaminated environments by triggering the transformation of As species by reduction, oxidation, methylation, demethylation, and cellular sequestration (summarized in [63]). In addition, *T. asperellum* can also remediate heavy-metal-contaminated environments. It was shown that *T. asperellum* protects crop health by reducing the bioavailability of heavy metals such as cadmium (Cd) and lead (Pb) through detoxification and sequestration [64]. The genomes of several *T. asperellum* strains have been sequenced (summarized in [63]).

In recent years, biopesticides have gained significant attention in the scientific world because they are a vital alternative to replacing the much-debated traditional chemical pesticides used in field crops. Secondary metabolites such as pyrone 6-pentyl-2H-pyran-2-one (6-pentyl-α-pyrone or 6-PP) play a significant role in the biological control of pests [65]. 6-PP is an unsaturated lactone with a molecular mass of 166 Da, first characterized by Collins and Halim (1972) [66] and identified as one of the key bioactive compounds of several *Trichoderma* species [65]. A strong relationship was found between the biosynthesis of this metabolite and the biocontrol ability of the producing species. 6-PP exhibits multiple actions, such as inhibiting mycelium growth, spore germination, and pigmentation of plant pathogenic fungi [67]. This metabolite also reduces the production of the mycotoxin deoxynivalenol by *Fusarium graminearum* and fusaric acid by *Fusarium moniliform* [68]. A large amount of 6-PP can be produced by *T. asperellum* using forced aeration on a solid-state fermentation system [65] for commercial production. Since 6-PP is a food-grade volatile metabolite, it can also be used post-harvest to protect storage crops.

In this research, we isolated an active ingredient (about 400 mg/liter) from the growth medium of *T. asperellum* chloroform extract that can completely inhibit (at a concentration of 8 mg per ml or more) the growth of *M. maydis.* The 6-PP has mild phytotoxicity to seed germination, and thus it should be further inspected in seedling and a full growing season to verify its safety for the plants. Achieving a clean and identified *T. asperellum* 6-PP-secreted product exhibiting high, potent antifungal activity against *M. maydis* is the first step in revealing its commercial potential as a new fungicide. Future studies should approve this purified component’s efficiency as a seed coating or other preventative treatments to protect highly susceptible maize cultivars against LWD as the second necessary step to this end. The current in vitro work must be followed by in vivo work that would include *M. maydis*-infected sprouts and mature plants under field conditions over an entire growth period. These additional steps, the sprouts and field experiments, will eventually enable the final decision—should the 6-PP be applied on a commercial field scale? If so, an approved and identified chemical or mixture of several materials with high efficiency against the LWD causal agent could be proposed as a new fungicide’s main ingredient and be commercially developed into a future product.

The soil fungus *T. longibrachiatum* (T7407) provided 29% more fungal inhibitory effects than *T. asperellum* in solid growth media (Figure 2). Active ingredients isolated from T7407 could likely be more potent. However, there is a risk of phototoxicity of these potential antifungal ingredients that should be taken under consideration. The impact of *T. longibrachiatum* Rifai (TL-9A) culture filtrate on the germination of different cultivated plants was tested by Celar et al. [69]. Compared to the control treatment, *T. longibrachiatum* culture filtrate significantly reduced the germination rate of onion seeds. On the other hand, the same culture filtrate significantly stimulated the initial germination of spinach, red beet, and chicory. This filtrate did not affect maize seed germination. Thus, in some plant species, this filtrate may have an opposite positive influence. For example, in spring barley seedlings, *T. longibrachiatum* 17 had no phytotoxic effect; on the contrary, this strain significantly increased the number of germinated seeds [70].

Still, potent phytotoxicity against seedling growth can exist, as found in the *Trichoderma harzianum* XS-20090075 strain (a coral-derived fungus) [71]. Indeed, newly isolated compounds from this fungus exhibited potent phytotoxicity against seedling growth of amaranth and lettuce. One of these compounds, Harzianone A (3) (molecular formula of C20H30O2), was similar to harzianone that was isolated from an alga-endophytic isolate of *T. longibrachiatum* [72]. Our *T. longibrachiatum* is also from a marine source, the sponge *Psammocinia* sp. [49].

As summarized by Zhao et al., 2019 [71], marine-derived *Trichoderma* spp. are a huge potential source for new phytotoxic compounds. They have been reported to represent a potential source for producing compounds with novel structures and remarkable bioactivities, such as trichodermamides A and B, dithioaspergillazine A, tandyukisins E, and F, as well as harzianone.

Thus, the phytotoxicity of *T. longibrachiatum* towards maize seedlings may depend on the source and strain of this *Trichoderma* species. The specific strain of *T. longibrachiatum* tested in this work was lately inspected for maize seedlings’ toxicity [26]. In potted, 47 days old plants, irrigation with *T. longibrachiatum* secreted metabolites marked a high value of seedlings emergence and shoot weight compared to the control. Such work during a full growth season, as well as isolating and identifying the active ingredient in the secreted metabolites of this species, are of great importance and should be included in follow-up work.

## 5. Conclusions

The maize (*Zea mays* L.) late wilt disease (LWD) caused by *Magnaporthiopsis maydis* is considered the most severe threat to commercial maize production in Israel. Various control strategies have been inspected over the years. The current scientific effort is focusing on eco-friendly approaches against the disease. The genus *Trichoderma*, a filamentous soil and plant root-associated fungi, is one of the essential biocontrol species, demonstrating over 60% of all the listed biocontrol agents used to reduce plant infectious diseases [73,74]. They produce different enzymes and elicit defense responses in plants, playing a significant role in biotic and abiotic stress tolerance, hyphal growth, and plant growth promotion [62]. *Trichoderma asperellum* was found to have biocontrol ability and protect crops against various phytopathogenic fungi, including the LWD agent. This research aimed at isolating and identifying *T. asperellum* secondary metabolites with antifungal action against *M. maydis*. From *T. asperellum* growth medium, the 6-PP secondary metabolite was isolated and identified with high potent antifungal activity against *M. maydis*. This metabolite previously exhibited in vivo and in vitro antifungal activities towards several plant pathogenic fungi [68]. Achieving clean and identified *T. asperellum* active ingredient(s)-secreted product(s) is the first step in revealing their commercial potential as new fungicides. Follow-up studies should test this component against the LWD pathogen in potted sprouts and in the field.

## Figures and Tables

**Figure 1 biology-10-00897-f001:**
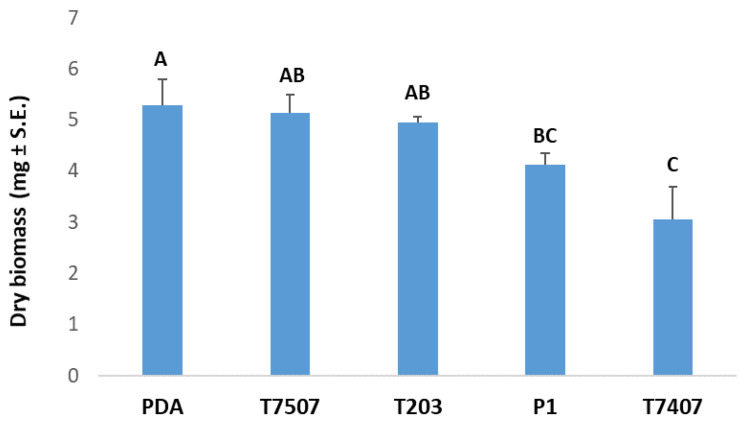
Effect of the growth fluid metabolites of *Trichoderma* isolates on the development of *M. maydis* submerged cultures. The *Trichoderma* isolates are *T. asperelloides* (T203), *T. longibrachiatum* (T7407, T7507), and *T. asperellum* (P1). All colonies were grown for six days. Control is potato dextrose broth (PDB) medium *M. maydis* cultures, maintained under the same conditions. The *M. maydis* dry weight was measured at the experiment’s end. Error lines represent a standard error of five repetitions. Statistical significance (*p* < 0.05) of variance was tested using the one-way ANOVA test and is represented by different letters (A–C) above the chart bars.

**Figure 2 biology-10-00897-f002:**
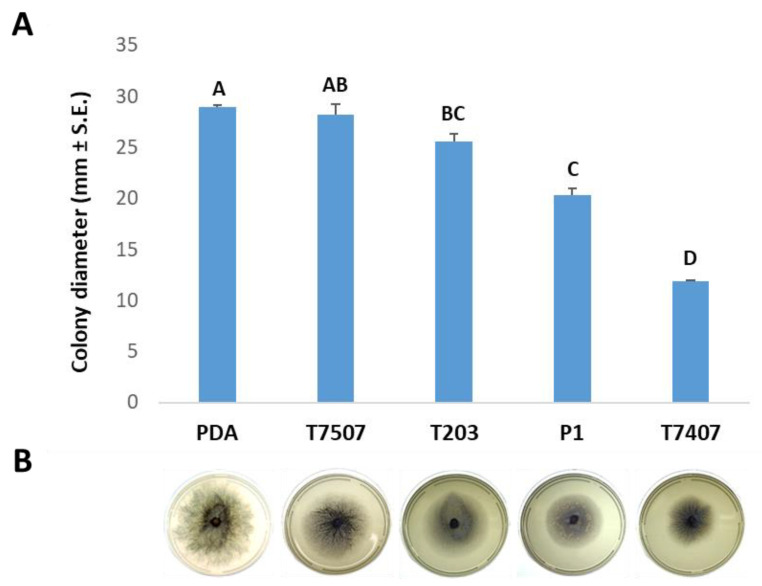
Effect of *Trichoderma* spp.-secreted metabolites on *M. maydis* solid media cultures. *Trichoderma* species tested are listed in Figure 1. The *Trichoderma* cultures’ filtrate was used to prepare potato dextrose agar (PDA) plates (instead of DDW). The plate was sown with *M. maydis* and incubated for six days. Control is PDA medium *M. maydis* cultures maintained under the same conditions. (**A**) Values are means from five biological replicates ± standard error. Different letters (A–D) indicate significant differences (one-way ANOVA test, *p* < 0.05). (**B**) Representative Petri dishes photograph of each treatment at the experiment’s end.

**Figure 3 biology-10-00897-f003:**
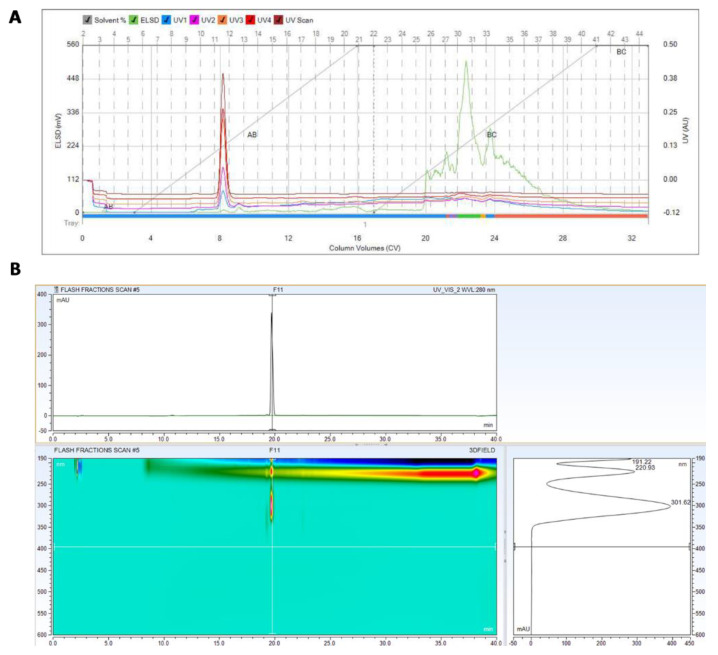
Purification of the *Trichoderma asperellum* (P1) active ingredient(s) that inhibit *M. maydis* growth. (**A**) The active compound(s) in the growth medium of *T. asperellum* were extracted using chloroform. The extracts’ fraction was purified using a flash chromatography system equipped with a flash silica column, diode array, and ELSD detectors. (**B**) HPLC chromatogram and UV spectra of the compound collected from fractions 11–12.

**Figure 4 biology-10-00897-f004:**
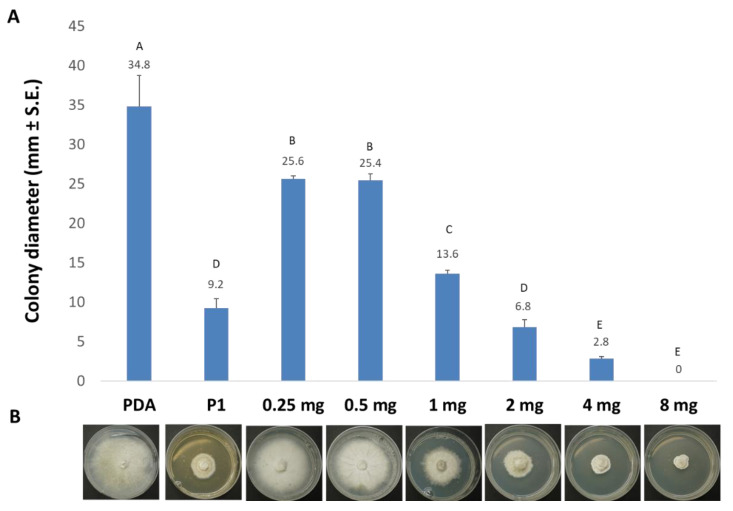
Bioassay of the purified *Trichoderma asperellum* (P1) active ingredient(s). (**A**) The purified active compound(s) in the growth medium of *T. asperellum* that inhibit *M. maydis* growth (described in Figure 3) were embedded in PDA at the indicated dosages. Thereafter, a plate *M. maydis* inhibition assay was performed for three days. Controls (on the left) are *M. maydis* grown on PDA and PDA embedded with P1 growth medium (crude) extracts. Vertical upper bars represent the standard error of the mean of four replicates. Levels not connected by the same letter (A–E) differ significantly from the other treatments in the same measure (*p* < 0.05, ANOVA). (**B**) Representative Petri dishes photograph of each treatment at the experiment’s end.

**Figure 5 biology-10-00897-f005:**
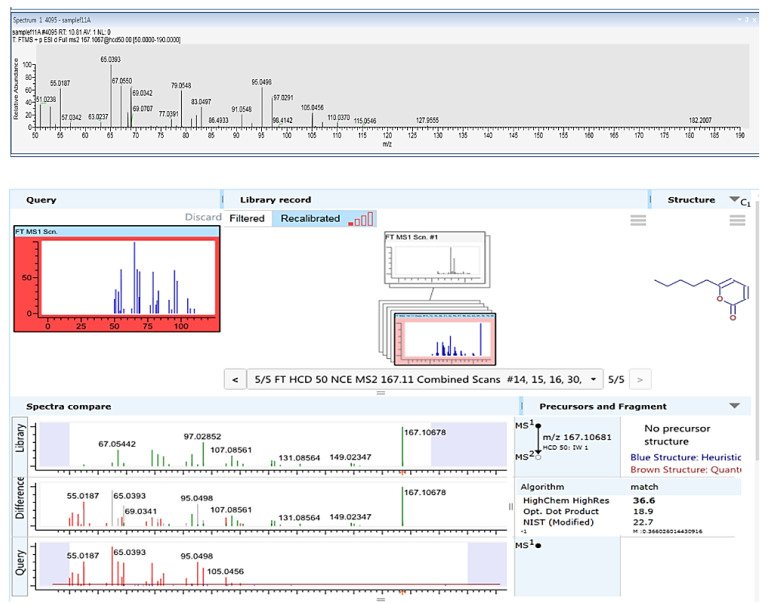
Identification of *Trichoderma asperellum* (P1) active ingredient(s) using LC–MS/MS analysis.

**Figure 6 biology-10-00897-f006:**
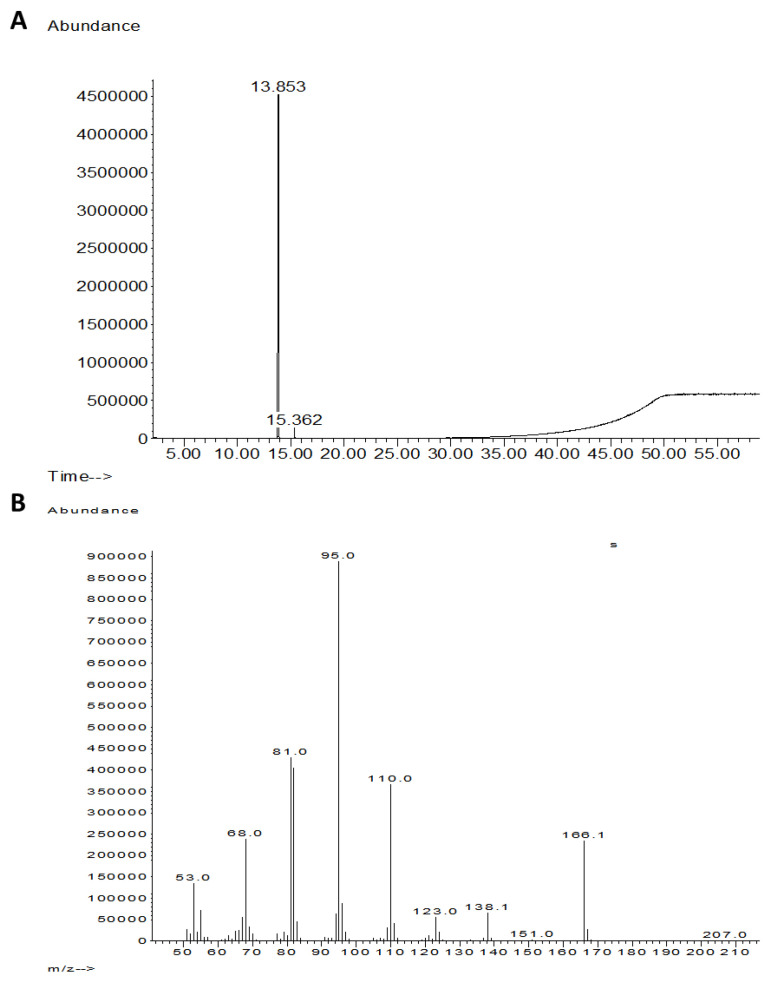
Identification of *Trichoderma asperellum* (P1) active ingredient(s) using GC-MS analysis. (**A**) GC–MS chromatogram and (**B**) MS spectra of the peak appearing at the chromatogram. Using the Willy library, the compound was identified as 6-Pentyl-α-pyrone (more than 95% fit between the spectra).

**Figure 7 biology-10-00897-f007:**
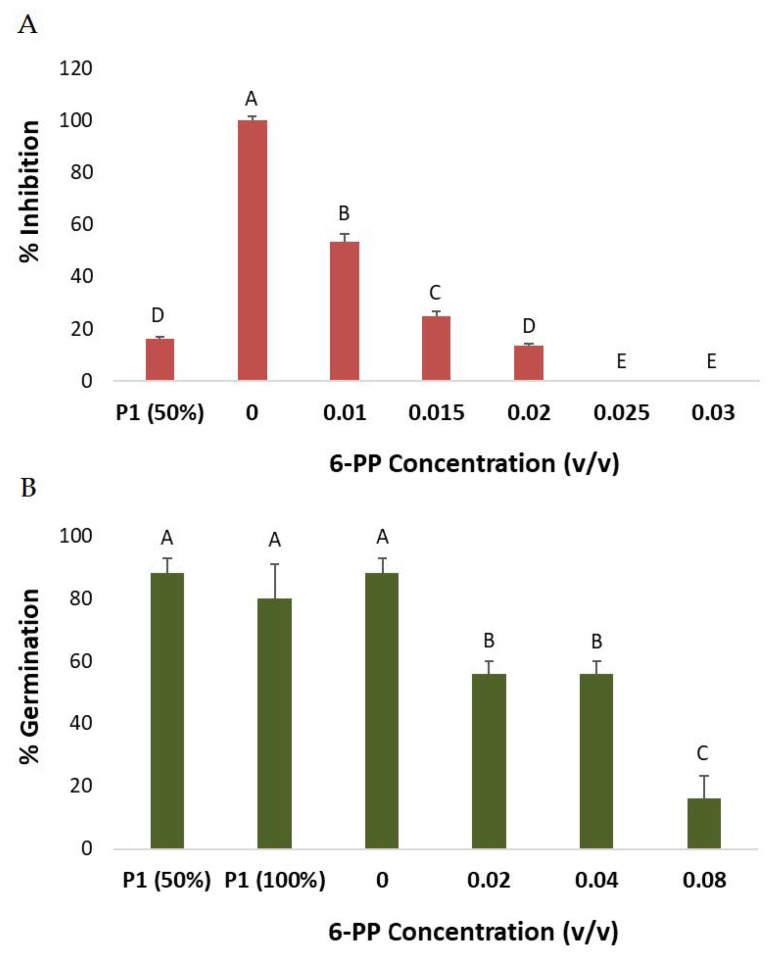
Seeds phytotoxicity assay for the 6-Pentyl-α-pyrone (6-PP). (**A**) *T. asperellum* (P1)-secreted metabolites (growth medium) or commercial 6-PP compound was embedded in PDA at the indicated dosages and tested for *M. maydis* inhibition, as described in Figure 2. (**B**) Impact of growth media of P1 or commercial 6-PP compound on corn seed germination. In each Petri dish, five grains were seeded in 3.5 mL of tap water, secretion products (growth medium filtrate after 6 days P1 growth), or commercial 6-PP compound at various concentrations, *v*/*v*). Data are average after three days of incubation. Vertical upper bars represent the standard error. Levels not connected by the same letter (A–E) differ significantly from the other treatments in the same measure (*p* < 0.05, ANOVA).

**Table 1 biology-10-00897-t001:** *Trichoderma* species used in this research.

Species	Designation	Origin	Reference	Confrontation Winner ^2^	Field Approval
*Trichoderma asperelloides*	T203	ATCC 36042, CBS 396.92	[26,50]	Antagonism	Yes
*Trichoderma longibrachiatum*	T7507	*Psammocinia* sp. ^1^	[49]	*T. longibrachiatum*	No
*Trichoderma longibrachiatum*	T7407	*Psammocinia* sp. ^1^	[26,49]	*T. longibrachiatum*	Yes
*Trichoderma asperellum*	P1	*Zea mays*, Prelude cv.	[27]	*T. asperellum*	Yes

^1^ Mediterranean sponge *Psammocinia* sp. ^2^
*M. maydis* confrontation assay results, including the following possibilities: *M. maydis* or *Trichoderma* sp. mycoparasitism (one of the fungi is growing above the colony surface of the other) and antagonism—none of the two fungi can extend above the other, and their growth was stopped in the meeting point with the other fungus, usually producing a dark line.

## Data Availability

The datasets generated during and/or analyzed during the current study are available from the corresponding author on reasonable request.

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
