# Peer review of "Trichoderma asperellum Secreted 6-Pentyl-α-Pyrone to Control Magnaporthiopsis maydis, the Maize Late Wilt Disease Agent"

_biology, 2021, doi:10.3390/biology10090897_

Round 1

Reviewer 1 Report

Late wilt disease (LWD) is a destructive disease and caused a significant threat to production on maize (Zea mays L.). It is a urgent need to explore biological agent against M. maydis. In this study, Three isolates of Trichoderma asperellum showed promising results, and a powerful (approx. 400 mg/L) active ingredient was isolated to control LWD, and identified it as 6-Pentyl-α-pyrone. This compound is a potential fungicide having high efficiency against the LWD causal agent. These results suitable for publish.

Author Response

A professional English scientific copy editor edited the entire manuscript. Still, some typo and wording mistakes may be found. We have made our best effort to correct them.

We thank the reviewer for investing time and effort in reviewing this manuscript. Your contribution is greatly appreciated.

Reviewer 2 Report

The manuscript entitled “Trichoderma asperellum secreted 6-Pentyl-α-pyrone to control 2 Magnaporthiopsis maydis, the maize late wilt disease agent” involves the isolation and identification of an active ingredient in the growth medium of Trichoderma asperellum (P1) with antifungal activity against M. maydis. The authors screened various Trichoderma isolates against the pathogen in liquid and solid growth media and they measured the dry weight parameter of target fungus and colony diameter, respectively, in the experiments. The authors found that P1 and T7407 possess the antifungal properties. Next, they moved with only P1 isolate and identified the active ingredient as 6-Pentyl-α-pyrone and further tested against M. maydis fungus. The active ingredient reduced the colony growth in dose-dependent manner.

The authors did a great job. They didn’t just identify the biocontrol agent; they also identified the active ingredient produced by biocontrol fungus which possess anti-plant pathogenic fungal properties. This study lays an important foundation for the development of ecofriendly pest management strategies. However, I have some major concerns regarding the current work listed below:

  1. The authors just pursued with P1, not with T7407. The justification was provided as “ asperellum (P1) is an endophyte isolated from grains of a maize genotype susceptible to LWD [23]. Thus, it is more likely that its secreted metabolites will have no phytotoxicity on the plants. For this reason, it was selected for the subsequent analytic work aimed at isolating and identifying the active ingredient(s) in its growth medium

Trichoderma longibrachiatum is a soil fungus. Are there any published reports on the phytotoxicity of this fungus?  Since T7407 provided 50% more fungal inhibitory effects compared to P1 in solid growth media. It is highly likely that active ingredients isolated from T7407 could be more potent.

  1. The second major concern is regarding the lack of strong evidence for the utility of 6-Pentyl-α-pyrone as anti-fungal compound. The authors tested the effects of this compound in vitro. To provide stronger evidence, they should have tested this compound on plants and test its efficacy against M. maydis. This kind of experiment will provide stronger evidence of using 6-Pentyl-α-pyrone as a biocontrol against maydis. Moreover, it will also test the authors hypothesis on lesser chances of phytotoxicity of P1.
  2. I think- the “hype” should be “hyphae” in sentences listed below-

Introduction-

“The pathogen is a soil-borne, hemibiotroph [3] and is seed-borne [4] spread as spores, sclerotia, or hype on plants’ remains”

About 10 days before harvest, the fungus hype and secreted materials block the plant’s water supply and lead to rapid de-44 hydration and death.

Author Response

Responses to Reviewer 2’s comments.

We want to express our sincere appreciation to the reviewer for the positive and constructive evaluations rendered to this manuscript. The time and effort invested are greatly appreciated and certainly contributed to the manuscript and improved it. Thank you.

The authors did a great job. They didn’t just identify the biocontrol agent; they also identified the active ingredient produced by biocontrol fungus, which possesses anti-plant pathogenic fungal properties. This study lays an important foundation for the development of eco-friendly pest management strategies. However, I have some major concerns regarding the current work listed below:

1. The authors just pursued with P1, not with T7407. The justification was provided as “ asperellum (P1) is an endophyte isolated from grains of a maize genotype susceptible to LWD [23]. Thus, it is more likely that its secreted metabolites will have no phytotoxicity on the plants. For this reason, it was selected for the subsequent analytic work aimed at isolating and identifying the active ingredient(s) in its growth medium.

Trichoderma longibrachiatum is a soil fungus. Are there any published reports on the phytotoxicity of this fungus?  Since T7407 provided 50% more fungal inhibitory effects compared to P1 in solid growth media. It is highly likely that active ingredients isolated from T7407 could be more potent.

Our T. longibrachiatum is from a marine source, the sponge Psammocinia sp. [1], but the reviewer is right; this point should be addressed and discussed in more detail. The following explanation was added to the Discussion (Lines 432-460):

“The soil fungus T. longibrachiatum (T7407) provided 29% more fungal inhibitory effects than T. asperellum in solid growth media. Active ingredients isolated from T7407 could likely be more potent. But, there is a risk of phototoxicity of these potential antifungal ingredients that should be taken under consideration. The impact of T. longibrachiatum Rifai (TL-9A) culture filtrate on germination of different cultivated plants was tested by Celar et al. [2]. Compared to the control treatment, T. longibrachiatum culture filtrate significantly reduced the germination rate of onion seeds. On the other hand, the same culture filtrate significantly stimulated the initial germination of spinach, red beet, and chicory. This filtrate did not affect maize seed germination. Thus in some plant species, this filtrate may have an opposite positive influence. For example, in spring barley seedlings, T. longibrachiatum 17 had no phytotoxic effect; on the contrary, This strain significantly increased the number of germinated seeds [3].

Still, potent phytotoxicity against seedling growth can exist, as found in the Trichoderma harzianum XS-20090075 strain (a coral-derived fungus) [4]. Indeed newly isolated compounds from this fungus exhibited potent phytotoxicity against seedling growth of amaranth and lettuce. One of these compounds, Harzianone A (3) (molecular formula of C20H30O2), was similar to harzianone that was isolated from an alga-endophytic isolate of T. longibrachiatum [5]. Our T. longibrachiatum is also from a marine source, the sponge Psammocinia sp. [1].

As summarized by Zhao et al., 2019 [4], Marine-derived Trichoderma spp. are a huge potential source for new phytotoxic compounds. They have been reported to represent a potential source for producing compounds with novel structures and remarkable bioactivities, such as trichodermamides A and B, dithioaspergillazine A, tandyukisins E, and F, as well as harzianone.

Thus, the phytotoxicity of T. longibrachiatum towards maize seedlings may depend on the source and strain of this Trichoderma species. As far as we know, the specific strain of T. longibrachiatum tested in this work was not yet inspected for maize seedlings’ toxicity. Such work, as well as isolating and identifying the active ingredients in this species secreted metabolites, are of great importance and should be included in follow-up work.”

2. The second major concern is regarding the lack of strong evidence for the utility of 6-Pentyl-α-pyrone as an antifungal compound. The authors tested the effects of this compound in vitro. To provide stronger evidence, they should have tested this compound on plants and test its efficacy against M. maydis. This kind of experiment will provide stronger evidence of using 6-Pentyl-α-pyrone as a biocontrol against maydis.

We agree. The current in vitro work must be followed by in vivo work that would include M. maydis infected sprouts and mature plants under field conditions over an entire growth period. These additional steps, the sprouts and field experiments, will eventually enable the final decision – is the 6-PP should be applied on a commercial field scale.

This explanation was embedded in the Discussion (lines 420-431):

“Achieving a clean and identified T. asperellum 6-PP secreted product exhibiting high potent antifungal activity against M. maydis is the first step in revealing its commercial potential as a new fungicide. Future studies should approve this purified component’s efficiency as a seed coating or other preventative treatments to protect highly susceptible maize cultivars against LWD as the second necessary step to this end. The current in vitro work must be followed by in vivo work that would include M. maydis infected sprouts and mature plants under field conditions over an entire growth period. These additional steps, the sprouts and field experiments, will eventually enable the final decision – is the 6-PP should be applied on a commercial field scale? If so, an approved and identified chemical or mixture of several materials having high efficiency against the LWD causal agent could be proposed as a new fungicide’s main ingredient and be commercially developed into a future product.”

We feel that this future work should be conducted and presented in a separate article. The current manuscript already contains many significant experimental results that should be given to encourage further follow-up research in the field.

A field trial with the 6-PP antifungal compound is indeed our next near-future goal, but it will require several months to accomplish it and will probably produce many new results. Combining the current manuscript with a field trial result will lead to a very long and loaded article. Therefore, we feel it is better to split the work into two steps and present them separately.

Moreover, it will also test the authors’ hypothesis on lesser chances of phytotoxicity of P1.

We conducted two new experiments, including maize seeds, to test our hypothesis on lesser chances of P1 phytotoxicity (new Figure 7). These new results were added to the manuscript:

  • Materials and Methods (lines 217-238):

2.4. Seeds phytotoxicity assay

The susceptible cultivar of sweet maize Prelude cv. from SRS snowy river seeds, Australia (supplied by Green 2000 Ltd., Israel) was chosen for the seed phytotoxicity test. This cultivar was tested previously and proved to be highly susceptible to late wilt [6-8]. To evaluate the secreted metabolite 6-Pentyl-α-pyrone (6-PP) phytotoxicity, we conducted a seed germination assay and compared the growth medium extract of T. asperellum (P1) to a commercial 6-PP compound (CAS Number: 27593-23-3, Sigma- Aldrich, Rehovot, Israel). The test included two steps. First, a plate inhibition assay was used as a bioassay to determine the antifungal activity of the P1 crude and the commercial 6-PP against M. maydis. To this end, the growth medium of P1 or commercial 6-PP compound was embedded in PDA at the indicated dosages and tested for M. maydis inhibition after three days, as described in section 2.2.

Second, selected concentrations of the commercial 6-PP were tested in seed germination assay against the P1 crude and a control (sterile tap water). Five seeds of the Prelude maize cultivar (not treated with traditional seed treatment fungicides or insecticides) were dipped in 1% (v/v) sodium hypochlorite for 3 min, washed in sterile double-distilled water (DDW), and then placed in a Petri dish with 3.5 ml of the inspected solution dissolved to the desired concentration in autoclaved DDW. We added five maize seeds to each Petri dish, and the plates were incubated at 28°C in the dark. Seeds germination percentage determination of all the seeds in each treatment were conducted three days later. A germinating seed was defined as a seed in which the seed coat was broken by the radicle. Untreated seeds (seeds soaked only in water were used as negative controls.”

Results (lines 342-362):

“To evaluate the secreted metabolite 6-PP phytotoxicity, we conducted a seed germination assay and compared the growth medium extract of T. asperellum (P1) to a commercial 6-PP compound. The test included two steps. First, a plate inhibition assay (Figure 7A) was used as a bioassay to determine the antifungal activity of the P1 crude and the commercial 6-PP against M. maydis. Fifty percent dilution of the P1 secreted metabolites had similar antifungal activity as the commercial compound’s 0.02% (v/v) concentration. Second, selected concentrations of the commercial 6-PP were tested in seed germination assay (Figure 7B) against the P1 crude. This assay revealed that the clean 6-PP (0.04%) was more phytotoxic than the crude (100%), which according to the plate bioassay, was expected to have similar antifungal activity. Still, at a relatively high concentration of clean 6-PP, the phytotoxicity at the sprouting phase was mild (36% inhibition compared to the control, p < 0.05).

(Figure 7)

Figure 7. Seed germination assay for the 6-Pentyl-α-pyrone (6-PP). A. T. asperellum (P1) secreted metabolites effect on M. maydis solid media cultures. The P1 growth medium or commercial 6-PP compound was embedded in PDA at the indicated dosages and tested for M. maydis inhibition, as described in Figure 2. B. Impact of growth media of P1 or commercial 6-PP compound on corn seed germination. In each Petri dish, five grins were seeded in 3.5 mL of tap water, secretion products (growth medium filtrate after 6 days P1 growth), or commercial 6-PP compound at various concentrations, v/v). Data are average after three days of incubation. Vertical upper bars represent the standard error. Levels not connected by the same letter (A-C) differ significantly from the other treatments in the same measure (p < 0.05, ANOVA).”

Discussion (lines 418-420):

“The 6-PP has relatively low phytotoxicity to seed germination, and thus, it is a promising candidate for seedling and full growth season LWD protection experiments.”

I think- the “hype” should be “hyphae” in sentences listed below:

Introduction - “The pathogen is a soil-borne, hemibiotroph [3] and is seed-borne [4] spread as spores, sclerotia, or hype on plants’ remains” About 10 days before harvest, the fungus hype and secreted materials block the plant’s water supply and lead to rapid dehydration and death.

The reviewer is correct. This is a typo — the word “hype” was replaced with “hyphae” in lines 34 and 44.

References

  1. Gal-Hemed, I.; Atanasova, L.; Komon-Zelazowska, M.; Druzhinina, I.S.; Viterbo, A.; Yarden, O. Marine isolates of Trichoderma spp. as potential halotolerant agents of biological control for arid-zone agriculture. Applied and environmental microbiology 2011, 77, 5100-5109.
  2. Celar, F.; Valic, N. Effects of Trichoderma spp. and Gliocladium roseum culture filtrates on seed germination of vegetables and maize. Journal of Plant Diseases and Protection 2005, 343-350.
  3. Sklyar, T.; Drehval, O.; Cherevach, N.; Matyukha, V.; Sudak, V.; Yaroshenko, S.; Kuragina, N.; Lykholat, Y.; Khromykh, N.; Didur, O. Antagonistic activity of microorganisms isolated from chernozem against plant pathogens. Ukrainian Journal of Ecology 2020, 10.
  4. Zhao, D.-L.; Yang, L.-J.; Shi, T.; Wang, C.-Y.; Shao, C.-L.; Wang, C.-Y. Potent phytotoxic harziane diterpenes from a soft coral-derived strain of the fungus Trichoderma harzianum XS-20090075. Scientific reports 2019, 9, 1-9.
  5. Miao, F.-P.; Liang, X.-R.; Yin, X.-L.; Wang, G.; Ji, N.-Y. Absolute configurations of unique harziane diterpenes from Trichoderma species. Organic letters 2012, 14, 3815-3817.
  6. Degani, O.; Dor, S.; Chen, A.; Orlov-Levin, V.; Stolov-Yosef, A.; Regev, D.; Rabinovitz, O. Molecular Tracking and Remote Sensing to Evaluate New Chemical Treatments Against the Maize Late Wilt Disease Causal Agent, Magnaporthiopsis maydis. J Fungi (Basel) 2020, 6, 54, doi:10.3390/jof6020054.
  7. Degani, O.; Movshowitz, D.; Dor, S.; Meerson, A.; Goldblat, Y.; Rabinovitz, O. Evaluating Azoxystrobin Seed Coating Against Maize Late Wilt Disease Using a Sensitive qPCR-Based Method. Plant Dis 2019, 103, 238-248, doi:10.1094/PDIS-05-18-0759-RE.
  8. Degani, O.; Regev, D.; Dor, S.; Rabinovitz, O.J.J.o.F. Soil Bioassay for Detecting Magnaporthiopsis maydis Infestation Using a Hyper Susceptible Maize Hybrid. 2020, 6, 107.

Reviewer 3 Report

Thank you, editor, for selecting me a potential reviewer for the manuscript entitled “Trichoderma asperellum secreted 6-Pentyl-α-pyrone to control Magnaporthiopsis maydis, the maize late wilt disease agent” which was submitted in “Biology”. I have found that manuscript has enough novelty to be publish in “Biology”. I would like to give some recommendations/suggestions which can increase the novelty of the manuscript.

I have read your abstract, it was good written, well write some more suggestions at the end of the abstract section.

Please note that keywords should not be repeated which already mentioned in the title, even some reviewers also suggested that keywords should not be a part of the abstract section. Please revise the following keywords: maize. Also, reduce the number of keywords, as 10 keywords are too many.

Line 30: Need reference of any previous study in your region.

It’s better to use scientific name of maize throughout the manuscript (just a suggestion).

Line 39: Which crop you are talking about? maize?

I have noticed that you have used this reference too many times: Degani, O.; Weinberg, T.; Graph, S. Chemical control of maize late wilt in the field. Phytoparasitica 2014, 42, 559-570, doi:10.1007/s12600-014-0394-5.

Your objectives are very simple, please write novelty of your study in more advance way.

Line 121: Write the author names also.

Results and discussions are well-written, please read your results at least once, as I found some minor grammar mistakes etc.

Well, this was very long experiment and author have made a lot of work on it, So I will definitely recommend this article for publication.

Best wishes

Author Response

Responses to Reviewer 3’s comments

We thank the reviewer for investing substantial efforts, which are undoubtedly contributing to this manuscript. The remarks and suggestions improved this paper’s scientific soundness and accurateness. Your contribution is greatly appreciated.

Thank you, Editor, for selecting me as a potential reviewer for the manuscript entitled “Trichoderma asperellum secreted 6-Pentyl-α-pyrone to control Magnaporthiopsis maydis, the maize late wilt disease agent,” which was submitted in “Biology.” I have found that the manuscript has enough novelty to be published in “Biology.” I would like to give some recommendations/suggestions which can increase the novelty of the manuscript.

Thank you for the positive and constructive evaluations rendered to this manuscript.

I have read your Abstract; it was good written, well write some more suggestions at the end of the abstract section. Please note that keywords should not be repeated which is already mentioned in the title; even some reviewers also suggested that keywords should not be a part of the abstract section. Please revise the following keywords: maize. Also, reduce the number of keywords, as 10 keywords are too many.

The reviewer is correct. The keywords were updated and now include 8 keywords that are not mentioned in the title:

biological control; Cephalosporium maydis; chromatography; crop protection; fungus; Harpophora maydis; microflora; mass spectrometry

Line 30: Need reference of any previous study in your region.

As suggested, we added a few more references of studies in our region.

It’s better to use the scientific name of maize throughout the manuscript (just a suggestion).

We used the scientific name of maize (Zea mays L.) at the beginning of the Abstract, the Introduction. We added the maize scientific name to the beginning of the Conclusion section, as well. Still, as customary in the scientific literature, we use the common name (maize or corn) to simplify the reading.

 Line 39: Which crop you are talking about? maize?

Yes, maize. We added the missing information, and the sentence now reads (lines 39-40): “According to Sabet et al. [2], the infection occurs during the first three weeks of maize growth.”

I have noticed that you have used this reference too many times: Degani, O.; Weinberg, T.; Graph, S. Chemical control of maize late wilt in the field. Phytoparasitica 2014, 42, 559-570, doi:10.1007/s12600-014-0394-5.

The reviewer is correct. We altered this reference with other references throughout the text. This reference is now cited in 2 places.

Your objectives are very simple. Please write the novelty of your study in a more advanced way.

We agree and update the objectives’ paragraph (at the end of the Introduction section), as follow (lines 113-121):

“The current research aimed at purifying and identifying ingredient(s) in the T. asperellum secreted metabolites with antifungal activity against the LWD pathogen. To this end, we screened various Trichoderma isolates against the pathogen in liquid and solid growth media. We measured the dry weight parameter of the target fungus and colony diameter, respectively. The secreted metabolites of T. asperellum (which show high antifungal properties against the LWD pathogen) were subjected to active ingredient isolation and identification using liquid and gas chromatography-mass spectrometry (LC-MS and GC-MS). Finally, the purified active ingredient was tested in M. maydis cultures dose-dependent growth inhibition assay and in maize seeds phytotoxicity assay.”

Line 121: Write the author names also.

The missing information was added, as suggested (lines 124-127):

“The M. maydis isolate Hm-2 (CBS 133165, CBS-KNAW Fungal Biodiversity Center, Utrecht, The Netherlands) was isolated by Amir Sharon (Tel Aviv University, Israel) from diseased maize plants collected from Sde Nehemia (Hula Valley, Upper Galilee, northern Israel) in 2001 and identified as previously described [1,2].”

Results and discussions are well-written. Please read your results at least once, as I found some minor grammar mistakes etc.

A professional English scientific copy editor edited the entire manuscript. Still, some typo and wording mistakes may be found. We have made our best effort to correct them.

Well, this was a very long experiment, and the authors have made a lot of work on it, So I will definitely recommend this article for publication.

Thank you.

References

  1. Degani, O.; Goldblat, Y. Ambient Stresses Regulate the Development of the Maize Late Wilt Causing Agent, Harpophora maydis. Agricultural Sciences 2014, 05, 571-582, doi:10.4236/as.2014.57060.
  2. Drori, R.; Sharon, A.; Goldberg, D.; Rabinovitz, O.; Levy, M.; Degani, O. Molecular diagnosis for Harpophora maydis, the cause of maize late wilt in Israel. Phytopathologia Mediterranea 2013, 52, 16-29.

Round 2

Reviewer 2 Report

The manuscript has been significantly improved. Congratulations to the authors for their great work and looking forward to their follow-up article. Since purified 6-PP showed phytotoxicity compared to crude extracts, the authors need to rephrase L418-420 “The 6-PP has relatively low phytotoxicity to seed germination, and thus, it is a promising candidate for seedling and full growth season LWD protection experiments.” What could be the possible reasons for that? Please mention that in the discussion.

Reviewer 3 Report

Author have made all the changes, therefore paper should be accepted in the current form.

Best wishes